# Drawing System with Dobot Magician Manipulator Based on Image Processing

**Pu-Sheng Tsai [1], Ter-Feng Wu [2],\*, Jen-Yang Chen [1] and Fu-Hsing Lee [3]**

[1] Department of Electronic Engineering, Ming Chuan University, Taoyuan 33348, Taiwan; pusheng@mail.mcu.edu.tw (P.-S.T.); jychen@mail.mcu.edu.tw (J.-Y.C.)
[2] Department of Electrical Engineering, National Ilan University, Yilan 26047, Taiwan
[3] Department of Electronic Engineering, China University of Science and Technology, Taipei 11581, Taiwan; irving.lee71024@gmail.com
\* Correspondence: tfwu@niu.edu.tw

**Abstract:** In this paper, the robot arm Dobot Magician and the Raspberry Pi development platform were used to integrate image processing and robot-arm drawing. For this system, the Python language built into Raspberry Pi was used as the working platform, and the real-time image stream collected by the camera was used to determine the contour pixel coordinates of image objects. We then performed gray-scale processing, image binarization, and edge detection. This paper proposes an edge-point sequential arrangement method, which arranges the edge pixel coordinates of each object in an orderly manner and places them in a set. Orderly arrangement means that the pixels in the set are arranged counterclockwise to the closed curve of the object shape. This arrangement simplifies the complexity of subsequent image processing and calculation of the drawing path. The number of closed curves represents the number of strokes in the drawing of the manipulator. In order to reduce the complexity of the drawing of the manipulator, a fewer number of closed curves will be necessary. To achieve this goal, we not only propose the 8-NN (abbreviation for eight-nearest-neighbor) search, but also use to the 16-NN search and the 24-NN search methods. Drawing path points are then converted into drawing coordinates for the Dobot Magician through the Raspberry Pi platform. The structural design of the Dobot reduces the complexity of the experiment, and its attitude and positioning control can be accurately carried out through the built-in API function or the underlying communication protocol, which is more suitable for drawing applications than other fixed-point manipulators. Experimental results show that the 24-NN search method can effectively reduce the number of closed curves and the number of strokes drawn by the manipulator.

**Keywords:** Dobot Magician manipulator; Raspberry; edge-point sequential arrangement; 8-NN search; 16-NN search; 24-NN search

## 1. Introduction

Artificial-intelligence robots offer promising market potential, particularly service-oriented and entertainment robots. These include the currently popular sweeping robot, the suitcase robot, the baby monitoring robot, and the housekeeper robot. Compared with functions such as medical diagnosis or automatic driving, learning and entertainment-oriented robots have a much higher degree of fault tolerance. Intelligent toys are another channel for firms to use the Internet-of-Things to enter the home market. In addition to providing domestic services, early education robots combine instructional functions with entertaining programs. A popular example is the Star Wars BB-8 SPHERO remote-control robot [1] and Anki's Cozmo social robot [2], shown in Figures 1 and 2, respectively,. Another example is the LEGO BOOST robot [3], which provides a building block kit for children to write their own programs, shown in Figure 3. Due to its comprehensive and flexible sporty performance, omni-directional mobile platforms are widely used in entertainment-oriented robots. In [4], Xu provide a structural analysis and motion resolving

solution to the platform based on four separate Mecanum wheels, building the perspective mathematical model in its kinematics and dynamics terms.

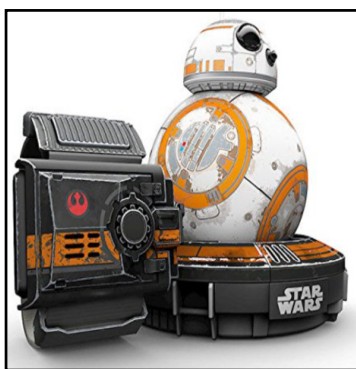

**Figure 1.** Star Wars BB-8 SPHERO [1].

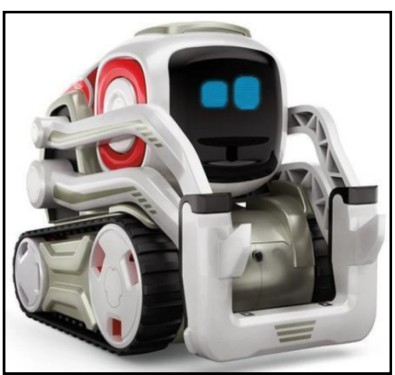

**Figure 2.** Anki's Cozmo [2].

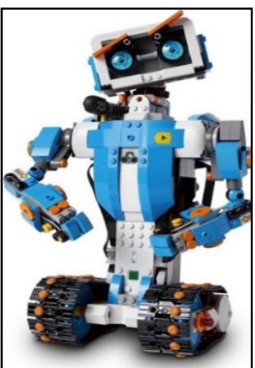

**Figure 3.** LEGO BOOST [5].

The range of applicability of artificial intelligence is diverse and continually expanding. In 1984, the performance robot Wabot-2 [6,7], shown in Figure 4, was published by the Research Office of Ichiro Kato, Waseda University, Japan. It can recognize music scores through its camera lens and play music flexibly with both hands and feet. In 2007, the Department of Mechanical Engineering of Zhongyuan University published the "Robot String Orchestra", shown in Figure 5. This system combines turning the piano, waving the bow, and pressing the string and kneading, and then cooperates with a multi-axis synchronous control string pressing device to realistically simulate the performance skills of human musicians [8,9].

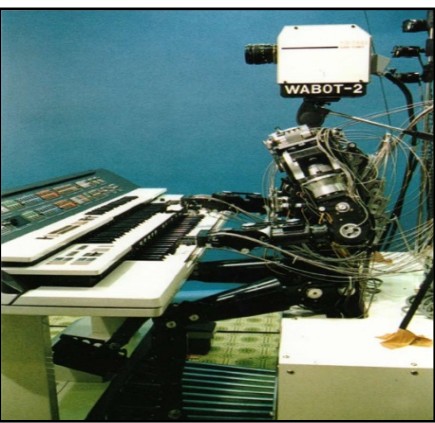

**Figure 4.** Performance robot of Waseda University [6].

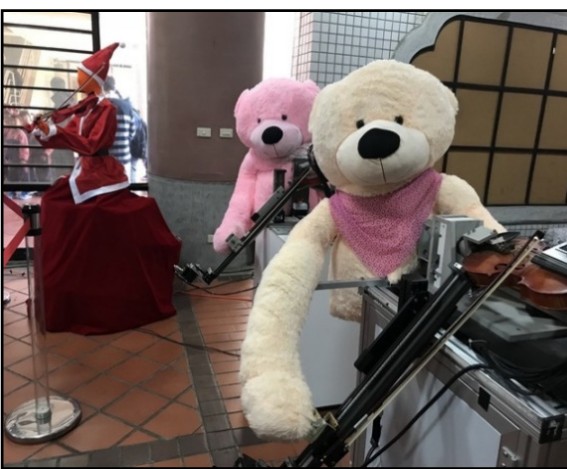

**Figure 5.** Robot string orchestra of Zhongyuan University [10].

Technological progress in artificial intelligence provides artists with new techniques, which is expected to ignite a revolution in visual art. British artists Nick and Rob Carter exhibited the "Dark Factory Portraits" in the London Gallery. These portraits were painted by an industrial robot named Kuka, which was developed for car production lines. The Carters' robot, Heidi, was adapted to paint portraits of well-known artists, as shown in Figure 6. The Heidi robot reads digital photos and uses image processing technology to convert two-dimensional (2D) pictures into corresponding code. This enables editing of each robot action, including drawing, selecting brushes, applying brushstrokes, and cleaning brushes. Robots represent tools for artists rather than substitutions for creative thinking and thus open up new possibilities for creative workers.

Many studies have investigated the use of robots for entertainment purposes. In [11], a robot was equipped with a powerful sensing ability, enabling it to draw a portrait on an uncalibrated surface of arbitrary shape. This achievement was aided by breakthrough developments in terms of mechanical arms. Robots have been applied to music composition, while developments related to 3D printing and mechanical engraving are opening up possibilities in the world of sculpture. Robots are also learning calligraphy [12], as shown in Figure 7. In this rule-bound art form, the robot can learn to write as fluently as a human. While font and pressing are easy to learn, the change of the shape of the pen hair and judgment of the dryness and wetness of the pen remain difficult for artificial intelligence. In recent years, artists and researchers have begun applying robots and automatic machines to develop novel forms of artistic painting. Interesting examples are given by the work of the system e-David, capable of controlling a variety of painting machines in order to create robotic artworks [13]. In [14], the non-photorealistic rendering techniques that are applied

together with a painting robot to realize artworks with original styles. Further examples of artistic robots are the interactive system for painting artworks by regions presented in [15] and the collaborative painting support system shown in [16].

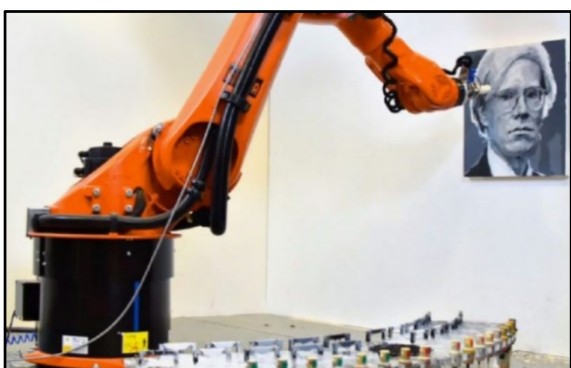

**Figure 6.** Portrait drawing by industrial robot Heidi [17].

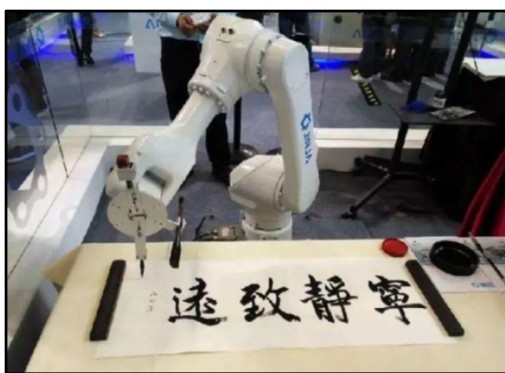

**Figure 7.** Calligraphy robot [18].

The Dobot Magician [19] is a desktop four-axis manipulator with an accuracy of 0.2 mm based on an STM32 industrial chip. It uses Python as the working platform and cooperates with the OpenCV image processing development kit and the Python API (the application interface) to control the manipulator. The Dobot Magician has been widely used in academic research [20]. For example, Tsao [21] wrote an Android application for smartphones to drive laser engraving [22]. Images are sent from the smartphone to Raspberry Pi, which calls the OpenCV function library to carry out image processing and find contour points. These are used to plan laser engraving path points, which are transmitted to the manipulator. In [23,24], Dobot and Raspberry Pi were applied to integrate image processing and robot-arm drawing. The plug-in camera streams the image, cooperates with the third-party kit OpenCV function library, carries out image processing, finds the contour points, classifies and calculates the drawing path points, and then transmits them to the manipulator for drawing. A similar set-up has been applied to perform electronic acupuncture and massages [25,26]. Acupoint recognition is achieved through image detection, and the manipulator is controlled to move to the corresponding acupoints. In [27], the Dobot Magician and machine vision were applied to the automatic optical detection of defects on production lines. The researchers in [28] integrated our atmega328p development board with the Dobot for electromechanical integration solid line automation. The DGAC design was proposed in [29], the spirit of gradient descent training is embedded in genetic algorithm (GA) to construct a main controller to search optimum control effort under possible occurrence of uncertainties. The effectiveness is demonstrated by simulation results, and its advantages are indicated in comparison with other GA control schemes for four-axis manipulators.

In this paper, the drawings of images and various curves are used to verify the feasibility of Dobot manipulator drawing system. Connected-component labeling (CCL) [30,31] is an algorithmic application of graph theory, where subsets of connected components are uniquely labeled based on a given heuristic which is used in computer vision to detect connected regions in binary digital images. Thus, in this paper, the method of CCL is applied to binary images construction which is used to cut an image in multiple single objects. Besides, to reduce the complexity of robot path planning, the edge-point sequential arrangement method is explored to rearrange pixels' image edge on the drawing path. This method is also investigated in this paper and is improved from the original idea of connected component labeling. A popular expression is the parametric B-spline curve, typically defined by a set of control points and a set of B-spline basis functions, which has several beneficial characteristics. It can be generalized as (a) it lies entirely within the convex hull of its control points; (b) it is invariant under a rotation and a translation of the coordinate axes; and (c) it can be locally refined without changing their global shape. The synthesis of the above concepts and B-spline properties leads to a successful methodology for trajectory planning application. In this paper, the divided B-spline function has been proposed by Tsai [32] which is used to describe the mathematical expressions of various curves.

## 2. System Architecture

In this paper, we aimed to realize the drawing function of the manipulator through machine vision. In addition to the mechanism of the four-axis manipulator, the Dobot Magician also includes expansion modules (gripper kit, air pump box, conveyor belt, photoelectric switch, and color recognition sensor), an external communication interface, and a control program. It also has 13 I/O extension interfaces and provides Qt, C#, Python, Java, VB, and other APIs for development. Image vision is carried out using a Raspberry Pi 3 external high-resolution USB camera. Raspberry Pi 3 adopts the Broadcom BCM2837 chipset, which is a four-core 64-bit ARMv8 Series CPU with a clock frequency of 1.4 GHz. It has a dual-core videocore IV 3D drawing core; provides 40 GPIO and 4 groups of USB 2.0; and supports 10/100 Ethernet, IEEE802.11 b/g/n wireless network, Bluetooth 4.1 network function, and 15 pin MIPI terminal camera connection function. The built-in Python language is used as the working platform. The contour pixel coordinates of the object in the image are transmitted to the Dobot through UART using gray-scale processing, image binarization, edge detection, and edge-point sequential arrangement to depict the edge contour of the object. The overall architecture consists of the following five units shown in Figure 8: Dobot Magician (including the writing-and-drawing module), Raspberry Pi 3 embedded microprocessor, camera, user interface, and image processing algorithm.

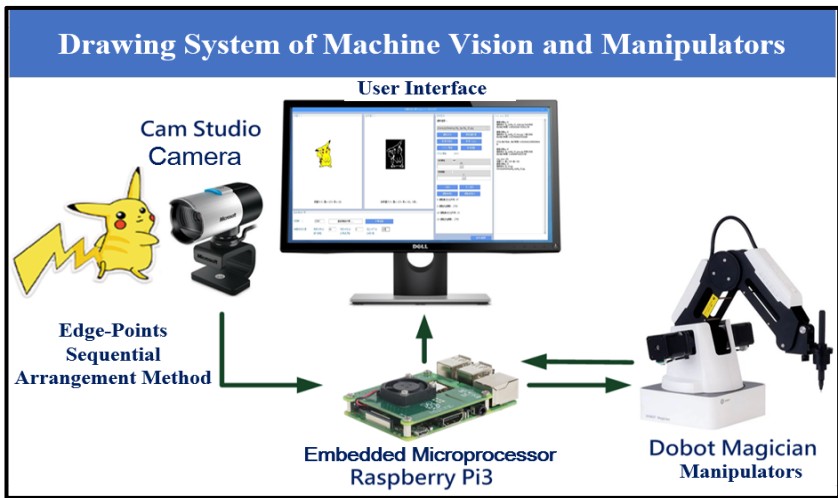

**Figure 8.** System architecture.

## 2.1. Specifications of Dobot Magician

As shown in Figure 9, the main body of the Dobot comprises the base, base arm, rear arm, forearm, and end effector. The corresponding four joints are defined as J1, J2, J3, and J4. We take J2 as the Cartesian origin (0,0,0), and the position of the end tool is defined as $(x, y, z)$. The power supply and warning lights are located above the base. The length of the forearm of the manipulator is 147 mm, the length of the rear arm is 135 mm, and the length of the base arm is 138 mm, as shown in Figure 10.

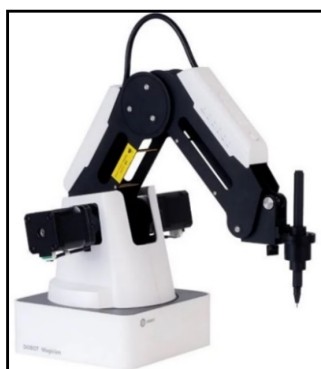

**Figure 9.** Image of Dobot.

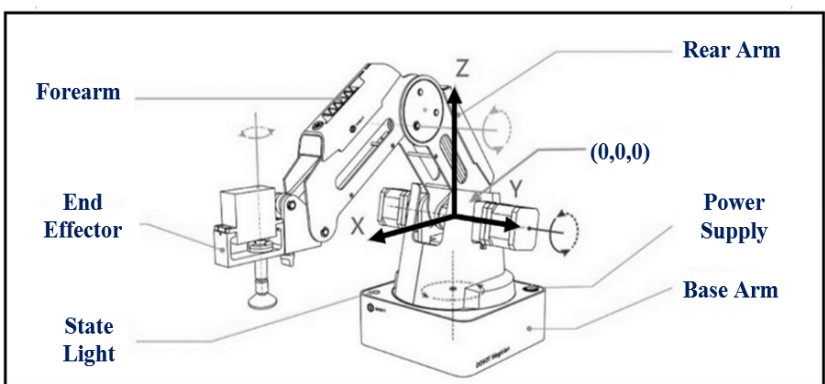

**Figure 10.** Diagram of Dobot structure.

The maximum load of the manipulator is 500 g, the maximum extension distance is 320 mm, the movement angle of the base arm joint J1 is ($-90$~$+90°$), the movement angle of the rear arm joint J2 is (0~$+85°$), the movement angle of the forearm joint J3 is ($-10$~$+90°$), and the movement angle of the end tool J4 is ($-135$~$+135°$). The working range of the *X-Y* axis of the manipulator is within the J1 activity angle and the maximum extension distance, while the activity space of the *Z* axis is within the J2 and J3 activity angle and the maximum extension distance. In addition, under a load of 250 g, the maximum rotation speed of the front and rear arms as well as the base is 320°/s, while the maximum rotation speed of the end tool is 320°/s.

## 2.2. Writing-and-Drawing Module

The writing-and-drawing module includes a pen and a pen holder, as shown in Figure 11. For our purposes, we replaced the pen with a brush. Figure 12 shows the specification parameters of Dobot with a pen-holder end tool.

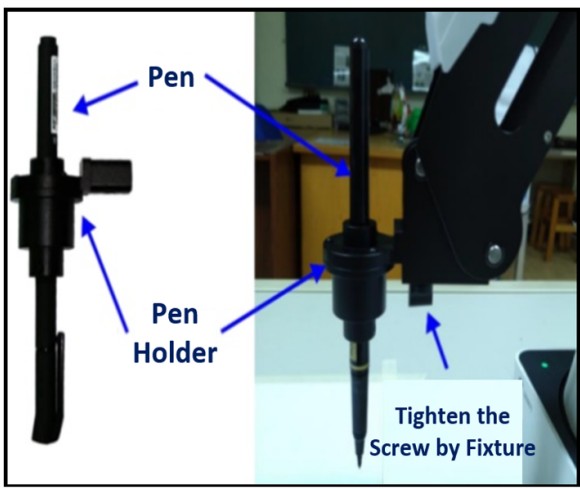

**Figure 11.** Writing and drawing kit.

| Specifications | Dobot Magician | | Pen holder end tool parameters |
|---|---|---|---|
| Payload | 500g | | |
| Max. Reach | 320mm | | Pen hole diameter : 10mm |
| Axis Movement | Joint 1 base | - 90°~ + 90° | |
| | Joint 2 rear arm | 0°~ + 85° | Suction cup end tool parameters |
| | Joint 3 forearm | - 10°~ + 90° | |
| | Joint 4 rotation servo | - 135°~ + 135° | Suction cup diameter : 20mm |
| Max Speed (250g workload) | Base, rear arm, forearm | 320°/s | |
| | Joint 4 rotation servo | 480°/s | Suction cup pressure : 35Kpa |
| Position Repeatability | 0.2mm | | |
| Power Supply | 100V~240V AC · 50/60Hz | | Claw end tool parameters |
| Power In | 12V/7A DC | | |
| Communication | USB · WIFI · Bluetooth | | Open/Close distance: 27.5mm |
| Extensible I/O Interface | I/O x 10 (Configurable as Analog Input or PWM Output) | | |
| Software | DobotStudio | | Drive: Pneumatic/8N |
| Working Temperature | - 10° ~ + 60°C | | |

**Figure 12.** Equipment specification parameters.

*2.3. Underlying Communication Protocol*

The API provided with the Dobot cannot directly call and control the manipulator on Raspberry Pi. Therefore, we used the Python pySerial module for serial communication to connect Raspberry Pi with the manipulator through the USB interface. The system must follow the communication protocol formulated by the Dobot. The pySerial module controls the manipulator through serial communication via the USB interface. The communication protocol includes the header, data length, transmitted data, and check code of the data packet. Table 1 shows the relevant communication protocol.

**Table 1.** Communication protocol of Dobot Magician.

| Header | Length | Payload | | | | Checksum |
|---|---|---|---|---|---|---|
| | | ID | Ctrl | | Params | |
| | | | rw | Is Queued | | |
| 0xAA 0xAA | 1 Byte | 1 Byte | High Byte 0x10 or 0x00 | Low Byte 0x00 or 0x01 | As Directed | Payload Calculate |

The physical layer is encoded by the IEEE-754 32-bit single precision floating-point number. For example, if we want to control the displacement of the manipulator from a certain point to the coordinate point ($x = 160$, $y = 90$, $z = -20$, $r = 0$). Then, the data encoding process is carried out as shown in Table 2.

**Table 2.** Example of signal encoding.

| Displacement coordinate | x | y | z | r |
|---|---|---|---|---|
| Decimal system | 160 | 90 | −20 | 0 |
| IEEE-754 | 0100001100100000 0000000000000000 | 0100001010110100 0000000000000000 | 1100000110100000 0000000000000000 | 0000000000000000 0000000000000000 |
| 16-Base system | 0x430x200x000x00 | 0x420xb40x000x00 | 0xc10xa00x000x00 | 0x000x000x000x00 |
| Small byte align type | 0x000x000x200x43 | 0x000x000xb40x42 | 0x000x000xa00xc1 | 0x000x000x000x00 |

$r$: the end tool is used when the steering gear (servo motor) is installed. Although this system is not used, the value must be brought in. Figure 13 shows the conversion results verified by the program written in Python; the following data can be used to control the manipulator with the PTP mode command through the write () method of pySerial.

```
>>> SetPTPCmd (PTPMode.PTPMOVLXYZINCMode, 160, 90, -20, 0)
Target Coordinate [x, y, z, r]= [160, 90, -20, 0]
SetPTPCmd Initial Coordinate= [170, 170, 19, 84, 16, 7] [160, 90, -20, 0] None
SetPTPCmd IEEE-754 HEX = [170, 170, 19, 84, 16, 7, 0, 0, 32, 67, 0, 0, 180, 66,
0, 0, 160, 193, 0, 0, 0, 0, 219]
PTPCmd Command :
=============================================================
Dobot Header= 0xaa, 0xaa,
Dobot Lenght= 0x13
Dobot ID= 0x54
Dobot CTRL= 0x10
Dobot PTPMode= 0x07
Dobot PRAMS= 0x07 0x00 0x00 0x20 0x43 0x00 0x00 0xb4 0x42 0x00 0x00 0xa0
0xc1 0x00 0x00 0x00 0x00
Dobot Check Sum= 0xdb
```

**Figure 13.** Verification of communication instructions.

*2.4. Camera Lens*

The image sources are files stored in the system or real-time images captured through the external CCD lens. The external lens has an automatic focusing function, as shown in Figure 8. It uses the transmission interface of USB 2.0 and is interconnected with the USB interface of Raspberry Pi3. Captured images are transmitted to the graphics window interface for image processing.

*2.5. Embedded Microprocessor*

Raspberry Pi offers powerful functionality, with a computing performance close to that of a small single-board computer and advantages including small size, low cost, low power consumption, and low noise. It can be connected to a keyboard, a mouse, or a display. This high, low-cost performance is achieved by using a 64-bit embedded microprocessor with an I/O entity pin to communicate with external sensors such as GPIO/I2C/UART interfaces. Specifications are as follows: (1) SOC: Broadcom BCM2837 chipset; (2) microprocessor: four-core 64-bit ARMv8 cortex-A8, 1.4 GHz; (3) display core: dual-core videocore IV 3D graphics core; (4) memory: LPDDR2, 1 GB; (5) network functions: 10/100 Ethernet, IEEE802.11 b/g/n wireless network, Bluetooth 4.1 (supporting general mode and low power consumption mode); (6) video output: HDMI (supporting Rev 1.3 and 1.4), composite video terminal (supporting NTSC and PAL) and 3.5 mm audio terminal; (7) USB: four groups of USB 2.0; (8) GPIO connection function: 40 pin 2.54 mm terminal, providing 27 GPIO and +3.3 V, +5 V, GND, and other power terminals; (9) camera connection function: 15 pin MIPI terminal (CSI-2); (10) display connection function: display serial interface terminal (DSI); (11) card reader: micro SD card reader, supporting SDIO; (12) operating system: boot with micro SD card, supporting Linux and Windows 10 IOT; and (13) size: 85 mm × 56 mm × 17 mm. Its appearance is shown in Figure 14.

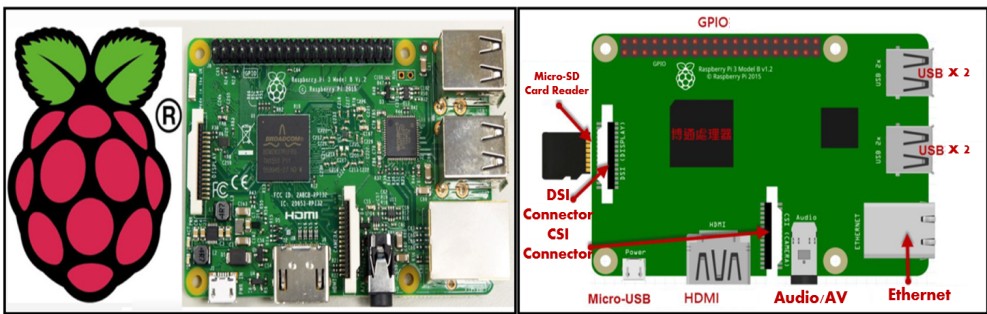

**Figure 14.** Components of Raspberry Pi.

## 3. Machine-Vision Image Processing

### 3.1. Image Preprocessing

The proposed drawing system mainly uses Python language and imports it into the OpenCV computer vision function library to read the image source. It compiles it into a visual window operation interface with the Tkinter graphical user interface function library to simplify the process of operating and controlling the manipulator. Before the robot begins to draw, the input image must be preprocessed to communicate through the underlying communication protocol of the robot.

As shown in Figure 15, the image source can be a pre-existing file in the system or a real-time image captured by the camera. Color images contain the pixel information of three primary colors: red (R), green (G), and blue (B). Color images must first be converted to grayscale to describe color brightness. Color pixels are converted to a gray-level value of 0–255. 0 represents black, and 255 is white. YIQ conversion is as follows:

$$\begin{bmatrix} Y \\ I \\ Q \end{bmatrix} = \begin{bmatrix} 0.299 & 0.587 & 0.114 \\ 0.596 & -0.275 & -0.321 \\ 0.212 & -0.528 & 0.311 \end{bmatrix} \begin{bmatrix} R \\ G \\ B \end{bmatrix} \tag{1}$$

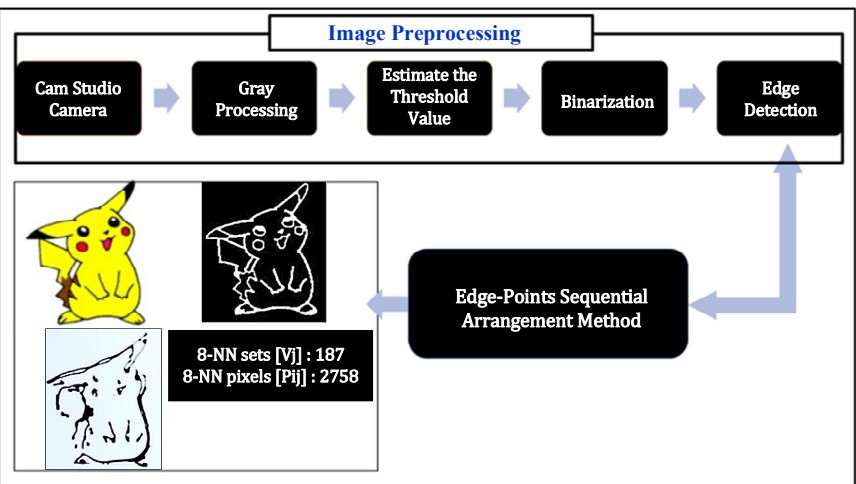

**Figure 15.** Flow chart of image processing.

Brightness signal $= 0.299 \times R + 0.587 \times G + 0.114 \times B$,

Therefore, $Y$ is the converted grayscale value and $I$, $Q$ are color components.

Image binarization uses trial and error or the Otsu algorithm to obtain the image threshold to distinguish whether each pixel belongs to black (0) or white (255). This helps to reduce noise. The grayscale values are represented by pixel points $f(x, y)$ combined

with the image threshold *k*, as depicted in Figure 16. Binary operation is then performed to generate a binary image output.

$$f(x,y) = \begin{cases} 0, & if \ f(x,y) \le k \\ 255, & otherwise \end{cases} \tag{2}$$

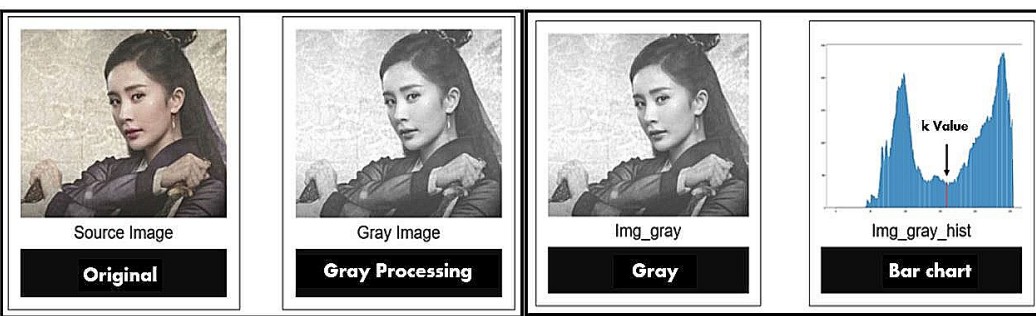

**Figure 16.** Image threshold *k*.

Therefore, $f(x,y) = 0$ indicates that the pixel value is black and $f(x,y) = 255$ indicates that the pixel value is white.

Sobel edge detection is used to find the edges of the object in the binary image. Two $3 \times 3$ horizontal and vertical matrix convolution kernels (as shown in Figure 17) or Sobel operators are established in advance. The edges of the image object are then obtained after convolution of each pixel in the image, as shown in Figure 18. Convolution is as follows:

$$g = \left[ G_x^2 + G_y^2 \right]^{\frac{1}{2}}$$
$$= \left\{ \left[ (z_3 + 2z_4 + z_5) - (z_1 + 2z_8 + z_7) \right]^2 + \left[ (z_7 + 2z_6 + z_5) - (z_1 + 2z_2 + z_3) \right]^2 \right\}^{\frac{1}{2}} \tag{3}$$

| $Z_1$ | $Z_8$ | $Z_7$ |
|---|---|---|
| $Z_2$ | $P_i^{\ j}$ | $Z_6$ |
| $Z_3$ | $Z_4$ | $Z_5$ |

| $-1$ | $-2$ | $-1$ |
|---|---|---|
| 0 | 0 | 0 |
| 1 | 2 | 1 |

| $-1$ | 0 | 1 |
|---|---|---|
| $-2$ | 0 | 2 |
| $-1$ | 0 | 1 |

**Figure 17.** $3 \times 3$ horizontal and vertical matrix convolution kernels.

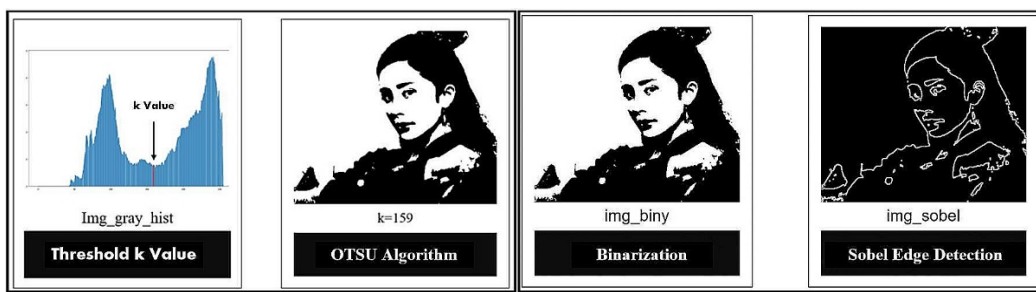

**Figure 18.** Results of Sobel edge detection.

Horizontal gradient after application of horizontal edge filter:

$$G_x = (z_3 + 2z_4 + z_5) - (z_1 + 2z_8 + z_7) \tag{4}$$

Horizontal gradient after application of vertical edge filter:

$$G_y = (z_7 + 2z_6 + z_5) - (z_1 + 2z_2 + z_3) \tag{5}$$

### 3.2. Edge-Point Sequential Arrangement

The purpose of the edge-point sequential arrangement is to arrange the pixel coordinates of each object edge point in the image into a set array. This is carried out in order to detect the adjacency and attributes between edge point pixels, ensuring that the edges of an object are placed in the same set. During image detection, the image is scanned from top to bottom and from left to right. A pixel content of "0" is defined as the background, and the foreground is denoted as "1". The scanning process involves object detection and edge detection. The number of objects and the number of edge coordinates of each object set are determined simultaneously. Here, the $i$th edge point coordinate contained in the $j$th object can be defined as $P_i^j = (x_i^j, y_i^j)$, where $j = 1, \cdots, m$, indicating that $m$ objects are detected in the image. And $i = 1, \cdots, n_j$, indicating that the number of $n_j$ edge points in the $j$th object. Therefore, the number of edge points contained in the image is $\sum_{j=1}^{m} n_j$. In the scanning process, once the object is detected, edge detection begins. For this, we apply a k-nearest-neighbor (k-NN) search and set k at 8, 16, and 24. Not every detected object will feature a complete closed curve, and there may be discontinuous breakpoints. If we apply only 8-NN, a complete curve cannot be obtained. Therefore, 16-NN and 24-NN expand the search to increase the allowable range of adjacency and matching attributes. As long as there are boundary points at adjacent positions, the algorithm will continue to search until a closed curve appears. Then, the edge point coordinates are recorded in the set array $V_j$. Figure 19 shows the 8-NN, 16-NN, and 24-NN search structures respectively. This paper takes the reverse clock as the search direction, and the search order is represented by numbers 1, 2, and 3.

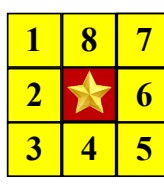 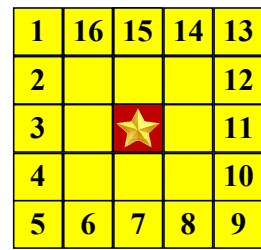 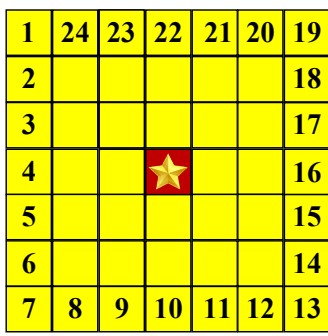

**Figure 19.** The 8-NN, 16-NN, and 24-NN search structures.

This algorithm searches the whole image to obtain the edge coordinate data of all objects in the image. The detailed procedure of the algorithm is described in the following:

Step 1: The operation procedure of the 8-NN search method is illustrated by the closed curve in Figure 20A, where the foreground pixels are denoted as "a", "b", "c" and so on. Start object detection from the first point in the image matrix. If an object is detected, proceed to Step 2. If a scanned pixel is background, no processing is performed. In this step, the $j$th scan to the foreground pixel is defined as the starting coordinate of the edge point of the $j$th object, expressed as $P_1^j$ and is recorded to the edge point set $V_j$. Referring to Figure 20A, where the first searched foreground coordinate is represented by a symbol "★".

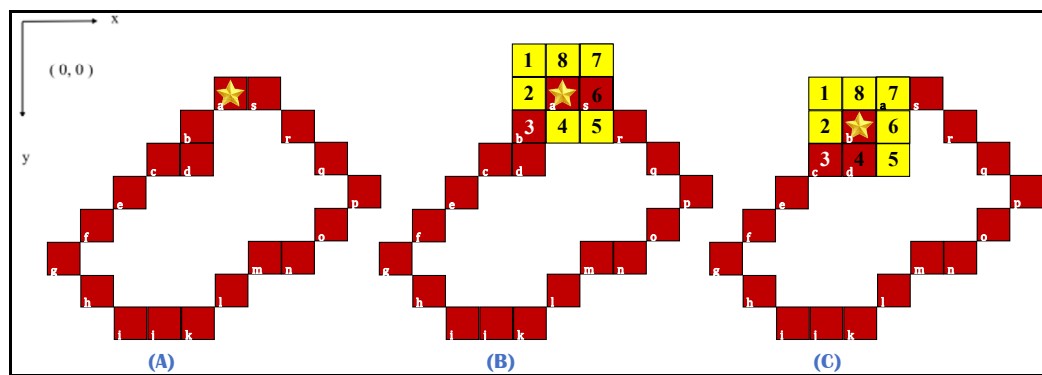

**Figure 20.** 8-NN search schematic diagram.

Step 2: With edge point $P_i^j$, $(i = 1)$ as the benchmark, perform 8-NN counterclockwise. The search method is to judge whether there are foreground pixels at eight points around "★" in the counterclockwise direction, as shown in Figure 20B. Once the foreground pixel "b" is scanned, the $i$ value is increased by 1, and the scanned edge point coordinates are stored in the set $V_j$.

Step 3: This pixel "b" is now regarded as the reference point of the next 8-NN scan, labeled as "★" and the previous pixel "a" is set to background "0". Repeat step 2, continue with the next scan, as shown in Figure 20C.

Step 4: If the 8-NN scan results in no detections, it may indicate a breakpoint, refer to Figure 21A. In this case, the 8-NN search seems to be invalid since the foreground pixels cannot be scanned, see Figure 21B. The 16-NN search method must be used to overcome the problem of breakpoints on closed curves, see Figure 21C. However, 8-NN search is still used for the next scan, and 16-NN search is used only when a breakpoint is encountered.

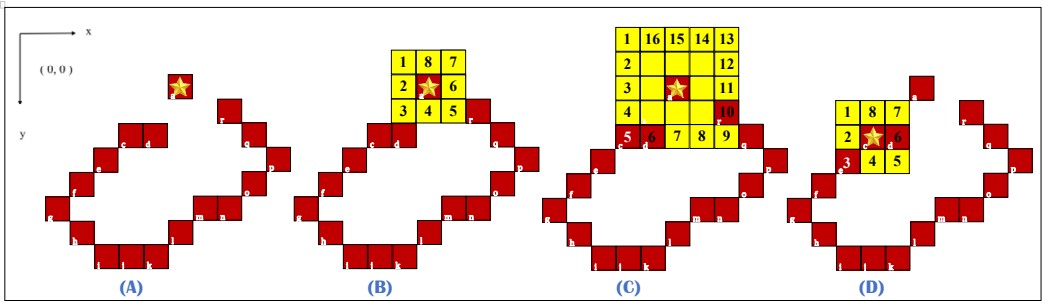

**Figure 21.** 16-NN search schematic diagram.

Step 5: In another extreme case, two breakpoints appear in the direction of the eight nearest neighbors of the benchmark, as shown in Figure 22A. At this time, not only 8-NN, but also the 16-NN search method is futile and cannot find the foreground pixel, see Figure 22B. If 16-NN returns no results, one should implement 24-NN. It can be found from Figure 22C that the foreground pixel "e" can be successfully searched by using 24-NN to solve the problem of two-layer breakpoints. Figure 22D illustrates that the next scan still adopts the 8-NN search with pixel E "e" as the reference point "★".

Step 6: Repeat Step 3. If the edge of the object is a closed curve, edge detection is complete when boundary point coordinates $P_i^j$ equal starting coordinates $P_1^j$; that is; $(x_i^j, y_i^j) = (x_1^j, y_1^j)$. If the edge of the object is not a closed curve, once all 24 of the nearest neighbors are found to be background pixels, scanning is stopped.

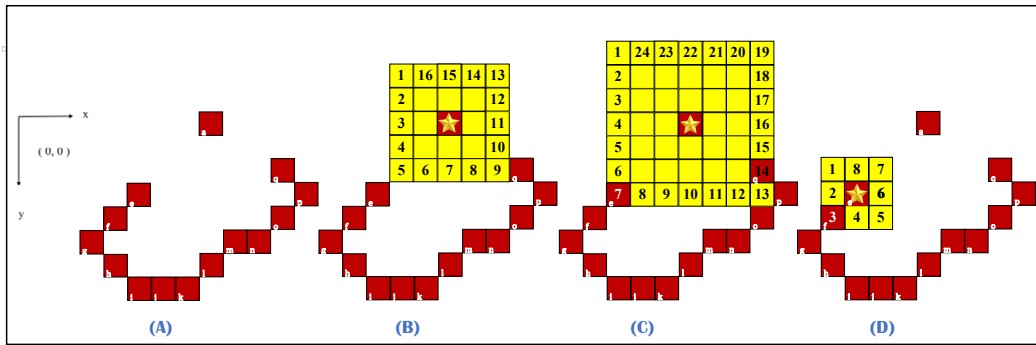

**Figure 22.** 24-NN search schematic diagram.

Step 7: Subtle noise exists in most images; therefore, a threshold value for the number of object edges is defined: $T_b$. If the number of edge pixels owned by an object $n_j$ is below the threshold, i.e., $n_j < T_b$, then treat object $j$ as noise. That is, delete object $j$ and its edge point records. Repeat Step 1 until all pixels in the image information are scanned; that is, the sequential arrangement of all edge points of all objects is complete. In this manner, we can find the number of objects in the image as well as the edge-point sequential coordinates of each object.

Let us consider the example presented in Figure 20B. The algorithm searches for edge-point coordinates in eight directions. When an edge point is detected in direction 3, the coordinates of this pixel are stored in the edge point set of the object. This pixel is then regarded as the reference point of the next 8-NN scan, and the initial pixel "★" is set as background data. When none of the eight nearest neighbors are foreground pixels, 16-NN is implemented. An edge point is then detected in the fifth neighbor, see Figure 21C. These coordinates are then stored in the edge point set of the object, and this pixel is regarded as the reference point of the next scan. The previous benchmark indicated by the orange edge point in Figure 21D is then considered background. When the 16-NN scan does not detect any foreground pixels, 24-NN is used to detect edge points. In Figure 22C, an edge point is detected at the seventh pixel. This is stored in the edge point set of the object. Pixel "e" is regarded as the reference point of the next scan, see Figure 22D.

### 3.3. Segmented B-Spline Curve

The spline curve has curve characteristics such as local control (Figure 23), convex hull property (Figure 24), partition of unity (Figure 25), and differentiability.

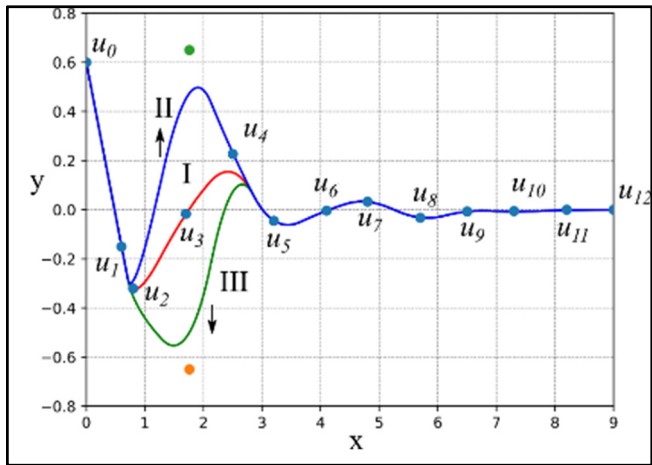

**Figure 23.** Schematic diagram of local control characteristics.

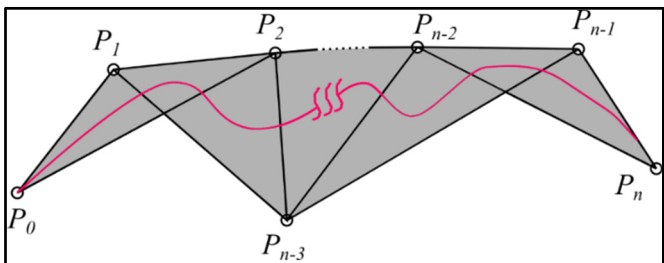

**Figure 24.** Schematic diagram of convex hull characteristics where $k = 2$.

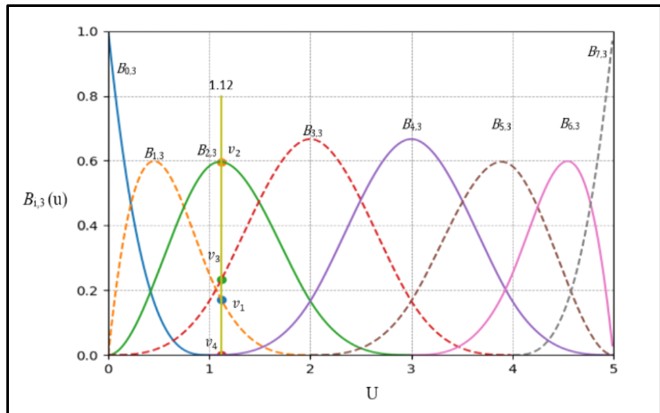

**Figure 25.** Unit division of B-spline curve.

The *B*-Spline base function can be defined by recursive deBoor-Cox equations:

$$B_{i,k}(u) = \frac{u - u_i}{u_{i+k} - u_i} B_{i,k-1}(u) + \frac{u_{i+k+1} - u}{u_{i+k+1} - u_{i+1}} B_{i+1,k-1}(u), \tag{6}$$

$$B_{i,0} = \left\{ \begin{array}{ll} 1 & if \quad u_i \leq u_{i+1} \\ 0 & otherwise \end{array} \right. , \quad and\ define\ \frac{0}{0} = 0. \tag{7}$$

.

The *B*-polynomial of the spline curve is defined as follows:

$$S(u) = \sum_{i=0}^{n} B_{i,k}(u) P_i \tag{8}$$

where node vector $U = \{u_0, u_1, \ldots, u_{n+k+1}\}$. The selection method is significantly related to the shape described by the spline curve. If the curve has the same control points, the shape of the curve will be different due to different node vectors.

Based on the above four characteristics, the element takes a uniform and open node vector, and the adjacent nodes are equidistant and have two ends. Tsai [32] suggested gradually disassembling the recursive formula into a combination of zero-order base functions. We then solve the corresponding coefficients to deduce Equation (9), which represents a matrix piecewise spline curve. This replaces the recursive expression of traditional deBoor equations [33,34]. $k = 3$ For example, where $S_j(t)$ represents the second $j$ Paragraph ($j = 0, 1, \ldots, m$). Each control point $P_j$, $P_{j+1}$, $P_{j+2}$, $P_{j+3}$ determines the shape of the curve, which is constructed by the third-order spline base function, the segment curve is shown in Figure 26. This base function has the smoothing characteristic of cubic differentiability, which is convenient for the trajectory design of constraint optimization.

The constructed spline curve becomes an input source of the drawing system, while the curve trajectory is fed back to the Dobot to test the drawing function of the manipulator.

$$
s_j(t) = \begin{bmatrix} P_j & P_{j+1} & P_{j+2} & P_{j+3} \end{bmatrix}
\begin{bmatrix}
-\frac{1}{6} & \frac{1}{2} & -\frac{1}{2} & \frac{1}{6} \\
\frac{1}{2} & -1 & 0 & \frac{2}{3} \\
-\frac{1}{2} & \frac{1}{2} & \frac{1}{2} & \frac{1}{6} \\
\frac{1}{6} & 0 & 0 & 0
\end{bmatrix}
\begin{bmatrix} t^3 \\ t^2 \\ t^1 \\ t^0 \end{bmatrix}
\tag{9}
$$

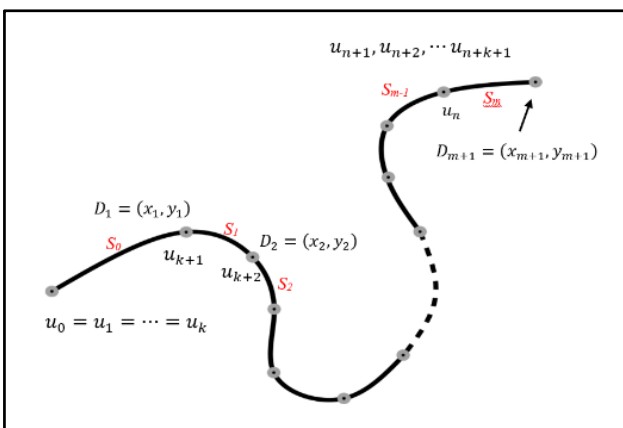

**Figure 26.** B-spline segment curve.

## 4. Experimental Results

To verify the feasibility of the proposed system, we performed functional tests with the proposed algorithms for edge-point sequential arrangement and piecewise spline curve representation. Figure 27 shows the module kit and measured environment loaded in the system.

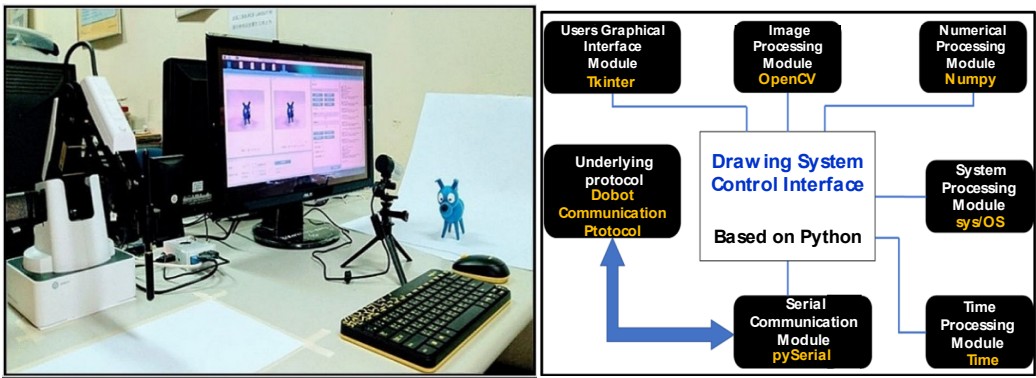

**Figure 27.** Loading of module kit and measured environment.

### 4.1. Graphical Window Interface

We imported the Tkinter GUI function library into the Python environment to write the graphical window interface. Image processing includes gray-scale processing, adjusting binarization thresholds, and edge detection. The results of image preprocessing can be seen in the image processing window. The function of robot drawing is designed through the robot control key and image processing display area. Based on the user interface written by Tkinter, the environment comprises the following five working blocks shown in Figure 28: (1) original image display area, (2) preprocessing display area, (3) function setting area, (4) edge-point sequence arrangement, and (5) robot moving coordinates.

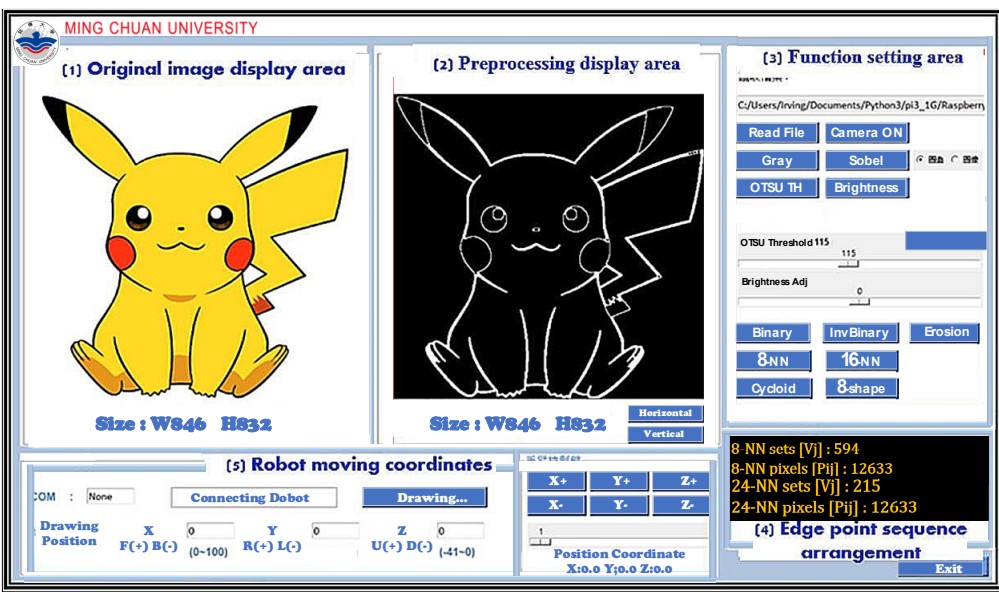

**Figure 28.** Graphical window interface.

In addition, as shown in Figure 29, the coordinate system of the image is usually different from the coordinate system defined by the manipulator. It is assumed that for an image $(C_x, C_y)$, the coordinate axis is $(gx, gy)$. The drawing range of the manipulator is limited to $(\triangle px, \triangle py) = (125 \text{ mm}, 230 \text{ mm})$, which means its coordinate axis is $(px, py)$. We define the starting point of the manipulator as $X$. The $X$-axis coordinate is then $offsetX$, the starting $Y$-axis coordinate is $offsetY$, and the image coordinates are expressed as follows:

$$px = \frac{\Delta px}{C_y} gy + 180 + offsetX,$$
$$py = \frac{\Delta py}{C_x} gx - 115 + offsetY \tag{10}$$

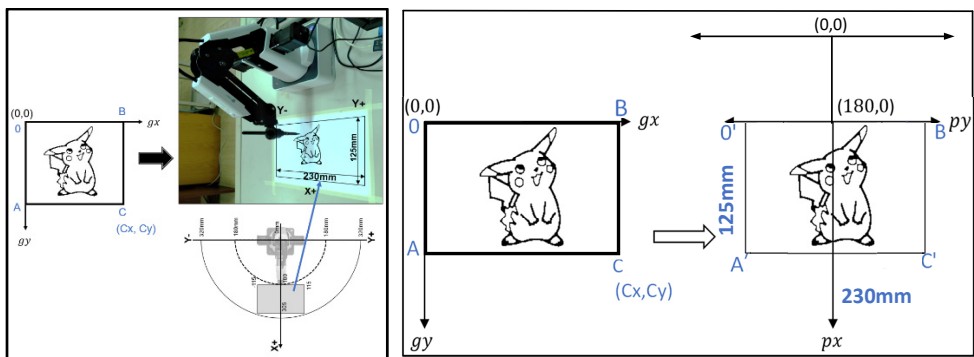

**Figure 29.** Coordinate conversion between image and manipulator.

*4.2. Mechanical Drawing of Edge-Point Sequential Arrangement*

After the image source is obtained, the number of closed sections in the film and the number of pixels in each closed section are obtained through image preprocessing and edge-point sequential arrangement. As shown in Figure 30, the image of Pikachu features 187 edge point sets obtained through an 8-NN search. Therefore, the robot arm must make 187 strokes to complete the image of Pikachu. For a 24-NN search, 56 edge point sets are obtained. For all k, 2758 pixels are searched. Once the system determines the closed curve and edge point coordinates, it encodes them into batch data according to the communication protocol formulated by the Dobot Magician. The pyserial module carries out serial communication through the USB interface to control the Dobot Magician. Figure 31 depicts the process of drawing Pikachu.

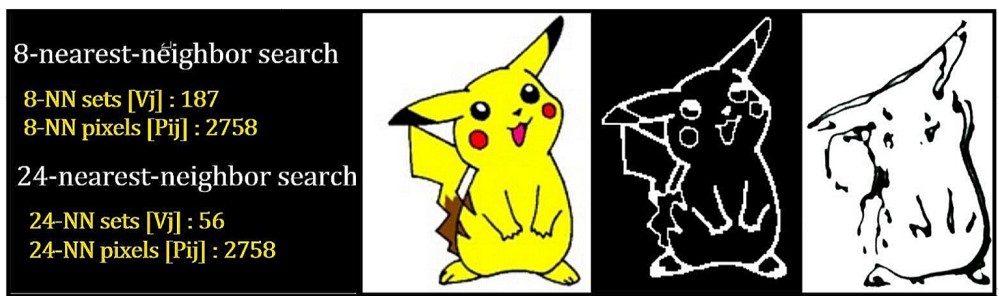

**Figure 30.** Results of drawing Pikachu.

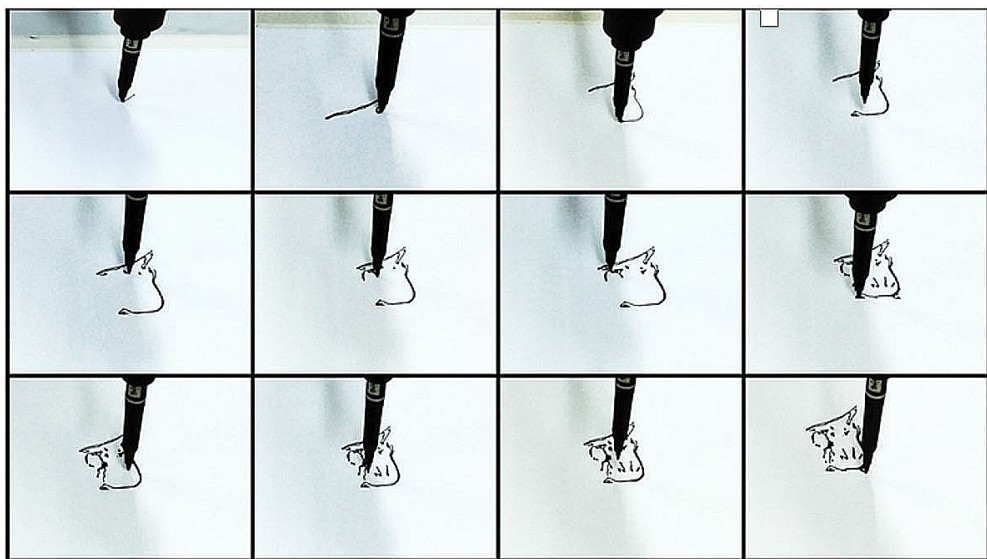

**Figure 31.** Drawing process of manipulator.

*4.3. Drawing of B-Spline Curve Manipulator*

According to Equation (9), the following two groups of control point coordinates are input: {(−10,10), (−5.135,12.194), (−1.1486,16.7401), (0,20), (2.1943,15.1351), (6.7491,11.1486), (10,10), (5.135,7.8057), (1.1486,3.2599), (0,0), (−2.1943,4.8650), (−6.7401,8.8514), (−10,10)} and {(−10,0), (−5,5), (0,0), (5,−5), (10,0), (5,5), (0,0), (−5,−5), (−10,0)}. Python then generates a group of four pointed cycloids and a group of eight-shaped spline curves. Figures 32–34 show the results of the two groups of curves drawn by the manipulator.

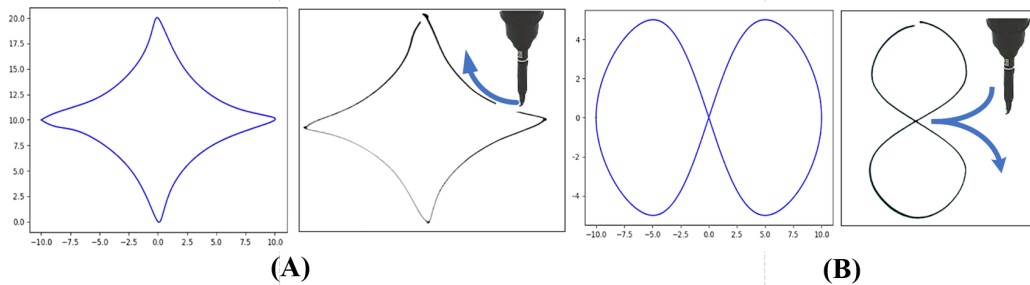

**Figure 32.** Drawing results of (**A**) quadrupole cycloid and (**B**) 8-shaped B-spline curve.

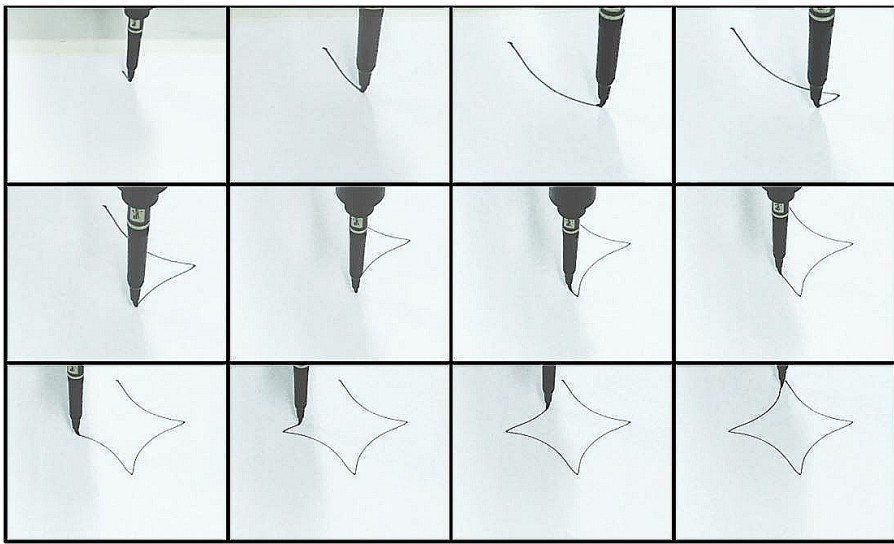

**Figure 33.** Drawing process of quadrupole cycloid manipulator.

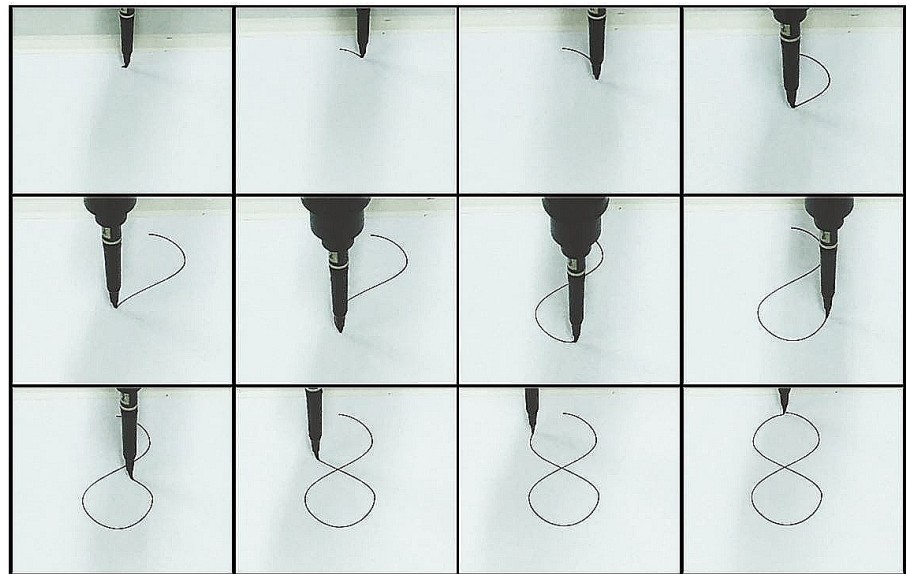

**Figure 34.** Drawing process of 8-shaped curve manipulator.

## 5. Conclusions and Directions for Future Research

In this paper, the Dobot Magician andf Raspberry Pi development platform are used to integrate image processing and robot-arm drawing. Since the image format is scanned from left to right and from top to bottom, the drawing manipulator cannot draw continuously. Our main contribution is an algorithm for edge-point sequential arrangement, which not only searches the closed curves in the drawn image but also arranges the edge pixel coordinates of each closed curve in order and places them in a set. This means that the pixels in the set are arranged in a direction counterclockwise to the closed curve. The number of closed curves is equal to the number of strokes drawn by the manipulator, which means that fewer closed curves reduce the complexity of the task. Thus, this paper proposes applying not only 8-NN but also 16- and 24-NN. Through the Raspberry Python platform, the drawing path points are converted into drawing coordinates for the robot arm. Experimental results show that 24-NN effectively reduces the number of closed curves and thus the number of strokes drawn by the manipulator. After forming a quadrupole cycloid and an eight-shaped closed spline curve through the divided B-spline curve algorithm, it establishes a group of array coordinate data for mechanical-drawing

simulation, verification, and comparison. The image processing of machine vision is closely related to the drawing effect of the manipulator. The effect of binarization can be adjusted using the threshold; however, edge detection is achieved by convoluting the horizontal and vertical edge filters. This effect depends entirely on the $3 \times 3$ arithmetic core. Directions for future research include improving the Sobel edge detection algorithm and using the proposed system with other end tools, such as laser engraving or 3D printing.

**Author Contributions:** P.-S.T. put forward the original idea of this paper and was responsible for the planning and implementation of the whole research plan. T.-F.W. is responsible for thesis writing and text editing. J.-Y.C. deduces the image processing algorithm in the manipulator drawing system and implements it with Python program. F.-H.L. establishes the hardware architecture and function verification of the manipulator drawing system. All authors have read and agreed to the published version of the manuscript.

**Funding:** This research was funded by Ministry of Science and Technology (MOST) (https://www.most.gov.tw, (accessed on 20 October 2021)), grant number MOST 110-2221-E-197-025.

**Institutional Review Board Statement:** Not applicable.

**Informed Consent Statement:** Not applicable.

**Data Availability Statement:** Not applicable.

**Acknowledgments:** The authors would like to thank the Ministry of Science and Technology (MOST) of the Republic of China, Taiwan, for financially supporting this research under Contract No. MOST 110-2221-E-197-025. (https://www.most.gov.tw, (accessed on 20 October 2021)).

**Conflicts of Interest:** The authors declare no conflict of interest.

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
