# Peer review of "Drawing System with Dobot Magician Manipulator Based on Image Processing"

_machines, doi:10.3390/machines9120302_

Round 1

Reviewer 1 Report

This paper proposed an edge-point sequential arrangement method in image processing part for a robot arm drawing system. On the whole, this work is complete and the method proposed may be effective. However, there are some issues as follow.

  1. In section 1 the authors reviewed the application of robot and the robot arm used in this paper but not mentioned the image processing algorithm concerned by this work.
  2. Image processing algorithms are traditional rather than intelligent. The edge-point sequential arrangement method proposed by authors is used to arrange the pixel coordinates of each object edge point in the image into a set array, which simplifies the complexity of subsequent image processing and calculation of the drawing path. From the perspective of the algorithm itself, it seems a little simple.
  3. To verify the proposed method, in section 4 the authors should give the comparative results which can be the processing time when using or not the proposed method.
  4. The authors can provide some theoretical analysis about the proposed algorithm.
  5. The arrangement of article’s structure needs to improve. Section 3.3 seems to have nothing to do with the image processing algorithm.

Reviewer 2 Report

In my opinion the work is interesting but it's not really clear how proposed work is really innovative respect to well known existing applications

Reviewer 3 Report

The paper presents a robotic drawing architecture based on a vision system and image processing techniques. The topics of the paper are interesting and suitable to the journal. However, the following points need to be improved before considering the paper for a journal publication.

1) The main contributions of the paper should be highlighted and clearly summarized. In particular, the novelties of the paper with respect to the present literature should be better discussed and commented.

2) The system architecture and the image processing algorithms implemented in this work should be better presented and illustrated.

3) It is not clear the role of the "machine vision" that is mentioned in the title of the paper.

4) The experimental results should be evaluated by means of performance metrics or quantitative indexes. It would be interesting to evaluate the performance of the drawing system by changing few software and/or hardware parameters.

5) More details on the path and trajectory planning for the robotic arm should be reported in the manuscript. It would be interesting to include the velocity and acceleration values used in the experimental tests.

6) The quality of the figures is overall very low for a journal publication. I suggest improving the whole style and quality of the manuscript. Where are Figures 1 to 8 taken from? The authors should cite the sources from where the images have been taken from.

7) English should be improved and the whole paper should be checked again for typos.

8) The literature review should be improved by adding some more recently published works on the topic. Some suggested references are:

Scalera, L., Seriani, S., Gasparetto, A., & Gallina, P. (2019). Non-photorealistic rendering techniques for artistic robotic painting. Robotics8(1), 10.

Gülzow, J. M., Paetzold, P., & Deussen, O. (2020). Recent developments regarding painting robots for research in automatic painting, artificial creativity, and machine learning. Applied Sciences10(10), 3396.

Guo, C., Bai, T., Lu, Y., Lin, Y., Xiong, G., Wang, X., & Wang, F. Y. (2020, August). Skywork-daVinci: A novel CPSS-based painting support system. In 2020 IEEE 16th International Conference on Automation Science and Engineering (CASE) (pp. 673-678). IEEE.

Round 2

Reviewer 1 Report

The author has replied to all the questions I raised.

Reviewer 3 Report

The paper has been improved with respect to the previous version.

Please clarify the meaning of "NN" in the abstract.
